# A Gradient Sampling Method
# With Complexity Guarantees for Lipschitz Functions
# in High and Low Dimensions[*]

**Damek Davis**
Cornell University
dsd95@cornell.edu

**Dmitriy Drusvyatskiy**
University of Washington, Seattle
ddrusv@uw.edu

**Yin Tat Lee**
University of Washington, Seattle
yintat@uw.edu

**Swati Padmanabhan**
University of Washington, Seattle
pswati@uw.edu

**Guanghao Ye**
Massachussetts Institute of Technology
ghye@mit.edu

## Abstract

Zhang, Lin, Jegelka, Sra, and Jadbabaie [1] introduce a method for minimizing Lipschitz functions with an efficiency guarantee of $O(\varepsilon^{-4})$. Their method is a novel modification of Goldstein's classical subgradient method. Their work, however, makes use of a nonstandard subgradient oracle model and requires the function to be directionally differentiable. Our *first* contribution in this paper is to show that both of these assumptions can be dropped by simply adding a small random perturbation in each step of their algorithm. The resulting method works on any Lipschitz function whose value and gradient can be evaluated at points of differentiability. Our *second* contribution is a new cutting plane algorithm that achieves better efficiency in low dimensions: $O(d\varepsilon^{-3})$ for Lipschitz functions and $O(d\varepsilon^{-2})$ for those that are weakly convex.

## 1 Introduction

The subgradient method [2] is a classical procedure for minimizing a nonsmooth Lipschitz function $f$ on $\mathbb{R}^d$. Starting from an initial iterate $x_0$, the method computes

$$x_{t+1} = x_t - \alpha_t v_t \text{ where } v_t \in \partial f(x_t). \tag{1}$$

Here, the positive sequence $\{\alpha_t\}_{t \geq 0}$ is user-specified, and the set $\partial f$ is the *Clarke subdifferential* [3, 4],

$$\partial f(x) = \operatorname{conv} \left\{ \lim_{i \to \infty} \nabla f(x_i) : x_i \to x, \ x_i \in \operatorname{dom}(\nabla f) \right\}.$$

In classical circumstances, the subdifferential reduces to familiar objects: for example, when $f$ is $C^1$-smooth at $x$, the subdifferential $\partial f(x)$ comprises of only the gradient $\nabla f(x)$, while for convex functions, it reduces to the subdifferential in the sense of convex analysis.

For functions $f$ that are weakly convex — a broad class of functions first introduced in English in [5] — the limit points $\bar{x}$ of the subgradient method are known to be first-order critical, meaning

---

[*]Authors ordered alphabetically.

36th Conference on Neural Information Processing Systems (NeurIPS 2022).

$0 \in \partial f(\bar{x})$. Recall that a function $f$ is called $\rho$-weakly convex if the quadratically perturbed function $x \mapsto f(x) + \frac{\rho}{2}\|x\|^2$ is convex. In particular, convex and smooth functions are weakly convex [6]. Going beyond asymptotic guarantees, finite-time complexity estimates are known for smooth, convex, or weakly convex problems [7–14].

Modern machine learning, however, has witnessed the emergence of problems far beyond the weakly convex problem class. Indeed, tremendous empirical success has been recently powered by industry-backed solvers, such as Google's TensorFlow and Facebook's PyTorch, which routinely train nonsmooth nonconvex deep networks via (stochastic) subgradient methods. Despite a vast body of work on the asymptotic convergence of subgradient methods for nonsmooth nonconvex problems [15–19], no finite-time nonasymptotic convergence rates were known outside the weakly convex setting until recently, with Zhang, Lin, Jegelka, Sra, and Jadbabaie [1] making a big leap forward towards this goal.

In particular, restricting themselves to the class of Lipschitz and directionally differentiable functions, [1] developed an efficient algorithm motivated by Goldstein's conceptual subgradient method [20]. Moreover, this was recently complemented by [21] with lower bounds for finding *near*-approximate-stationary points for nonconvex nonsmooth functions.

While a significant breakthrough in both result and technique, one crucial limitation of [1] is that their complexity guarantees and algorithm use a nonstandard first-order oracle whose validity is unclear in examples. To elaborate, their algorithm requires the following oracle access: given $x, u \in \mathbb{R}^d$ solve the *auxiliary convex feasibility problem*:

$$\text{find } g \in \partial f(x) \text{ subject to } \langle g, u \rangle = f'(x, u). \tag{2}$$

The first issue with this oracle is that no general recipe exists for representing the full subdifferential $\partial f(x)$ analytically, and evaluating even an arbitrary element of the subdifferential can be highly non-trivial [22, 23]. Moreover, $\partial f(x)$ could be a very complicated set, e.g., for a deep ReLU neural network, the subdifferential is a polyhedron with a potentially huge number of facets, making the complexity of (2) unclear.

Further, [1] claim that for a composition of directionally differentiable functions with a closed-form directional derivative for each function, we can find the desired $g$ by the chain rule. While the chain rule does compute the directional derivative $f'(x, u)$, to the best of our knowledge, this does not translate to solving (2). This is owing to the crucial fact that the chain rule (and sum rule) can easily fail for the computation of the subdifferential[2] (although these are indeed valid for the directional derivative of a composition of directionally differentiable functions). We believe that this could potentially render the oracle of [1] computationally intractable.

Finally, we are unaware of other optimization algorithms imposing this oracle model. Therefore, at face value, the convergence guarantees of [1] are not comparable to those of others.

## 1.1 Our results

**Weakly convex optimization via a standard oracle.** Our first contribution is to recover the complexity result of [1] under a much weaker assumption: specifically, we *replace the non-standard assumption in (2) with a standard first-order oracle model*. We show (Section 2) that a simple, yet critical, modification of the algorithm of [1], wherein one simply adds a small random perturbation in each iteration, works for any Lipschitz function assuming only an oracle that can compute gradients and function values at almost every point of $\mathbb{R}^d$ in the sense of Lebesgue measure. In particular, such oracles arise from automatic differentiation schemes routinely used in deep learning [24, 19]. Our end result is a randomized algorithm for minimizing any $L$-Lipschitz function that outputs a $(\delta, \epsilon)$-stationary point (Definition 1) after using at most $\widetilde{\mathcal{O}}\left(\frac{\Delta L^2}{\epsilon^3 \delta} \log(1/\gamma)\right)$ [3] gradient and function evaluations. Here $\Delta$ is the initial function gap and $\gamma$ is the failure probability.

---

[2]We provide a simple example to demonstrate this claim: Consider the function $f(x, y) = f_1(x, y) + f_2(x, y)$ with $f_1(x, y) = |x|$ and $f_2(x, y) = -|x|$. Choose the direction $u = (0, 1)$, and let $z = (0, 0)$. Then, $f_1'(z, u) = f_2'(z, u) = 0$, and $\partial f_1(z) = \partial f_2(z) = [-1, 1] \times 0$. Therefore, to satisfy the oracle (2), for $f_1$, we may choose the subgradient $v_1 = (-1, 0)$, and for $f_2$, we may choose the subgradient $v_2 = (-1, 0)$ since $\langle v_1, u \rangle = 0 = \langle v_2, u \rangle$. However, $v_1 + v_2 = (-2, 0)$, which is not a subgradient of $f = 0$ at $z$.

[3]Throughout the paper, we use $\widetilde{\mathcal{O}}(\,\cdot\,)$ to hide poly-logarithmic factors in $L, \delta, \Delta$, and $\epsilon$.

In light of the above modifications, our algorithm is implementable in the many important settings like deep neural networks that [1] is not. Along the way, we also simplify their proof techniques by providing a geometric viewpoint of the algorithm.

We would like to highlight the concurrent work of Tian, Zhou, and Man-Cho So [25], which obtains this part of our result with a very similar technique.

**Improved complexity in low dimensions.** Having obtained the result of [1] within the standard first-order oracle model, we then proceed to investigate the following question.

*Can we improve the efficiency of the algorithm in **low dimensions**?*

In addition to being natural from the viewpoint of complexity theory, this question is well-grounded in applications. For instance, numerous problems in control theory involve minimization of highly irregular functions of a small number of variables. We refer the reader to the survey [26, Section 6] for an extensive list of examples, including Chebyshev approximation by exponential sums, spectral and pseudospectral abscissa minimization, maximization of the "distance to instability", and fixed-order controller design by static output feedback. We note that for many of these problems, the gradient sampling method of [26] is often used. Despite its ubiquity in applications, the gradient sampling method does not have finite-time efficiency guarantees. The algorithms we present here offer an alternative approach with a complete complexity theory.

The second contribution of our paper is *an affirmative answer to the highlighted question*. We present a novel algorithm that uses $\widetilde{\mathcal{O}}\left(\frac{\Delta L d}{\epsilon^2 \delta} \log(1/\gamma)\right)$ calls to our (weaker) oracle. Thus we are able to trade off the factor $L\epsilon^{-1}$ with $d$. Further, if the function is $\rho$-weakly convex, the complexity improves to $\widetilde{\mathcal{O}}\left(\frac{\Delta d}{\epsilon \delta} \log(\rho)\right)$, which matches the complexity in $\delta = \epsilon$ of gradient descent for smooth minimization. Strikingly, the dependence on the weak convexity constant $\rho$ is only logarithmic.

To put this contribution in perspective, assume for now $\delta = \epsilon$: then, our algorithm's dependence on $\epsilon$ in the case of Lipschitz, weakly convex functions is likely optimal in low dimensions, following a conjecture by Bubeck and Mikulincer [27] on the optimality of gradient descent for *smooth optimization* in dimension $d = \log(\frac{1}{\epsilon})$ (thus matching the lower bound by Carmon, Duchi, Hinder, and Sidford [28]). Aside from possible optimality, the logarithmic dependence on smoothness/weak convexity exhibited by our iteration complexity is a significant improvement over the prior result of either $O(1/\epsilon^4)$ by [1] or Nemirovski and Yudin's rate of $O(1/\epsilon^2)$ with a polynomial dependence on smoothness. In the process, we also show that the minimal-norm element of the Goldstein-subdifferential in low dimensions can be found in time $O(\log(1/\epsilon))$, thus settling a question open since the 70s.

**Techniques.** The main idea underlying our improved dependence on $\epsilon$ in low dimensions is outlined next. The algorithm of [1] comprises of an outer loop with $\mathcal{O}\left(\frac{\Delta}{\epsilon \delta}\right)$ iterations, each performing either a decrease in the function value or an ingenious random sampling step to update the descent direction. Our observation, central to improving the $\varepsilon$ dependence, is that the violation of the descent condition can be transformed into a gradient oracle for the problem of finding a minimal norm element of the Goldstein subdifferential. This gradient oracle may then be used within a cutting plane method, which achieves better $\varepsilon$ dependence at the price of a dimension factor (Section 3).

**Limitations.** One limitation of our work is that our second contribution does not immediately extend to the stochastic setting. We consider this to be an interesting open problem to resolve.

**Notation.** Throughout, we let $\mathbb{R}^d$ denote a $d$-dimensional Euclidean space equipped with a dot product $\langle \cdot, \cdot \rangle$ and the Euclidean norm $\|x\|_2 = \sqrt{\langle x, x \rangle}$. The symbol $\mathbb{B}_r(x)$ denotes an open Euclidean ball of radius $r > 0$ around a point $x$. Throughout, we fix a function $f \colon \mathbb{R}^d \to \mathbb{R}$ that is $L$-Lipschitz, and let $\mathrm{dom}(\nabla f)$ denote the set of points where $f$ is differentiable—a full Lebesgue measure set by Rademacher's theorem. The symbol $f'(x, u) \overset{\text{def}}{=} \lim_{\tau \downarrow 0} \tau^{-1}(f(x + \tau u) - f(x))$ denotes the directional derivative of $f$ at $x$ in direction $u$, whenever the limit exists.

## 2 Interpolated normalized gradient descent

In this section, we describe the results in [1] and our modified subgradient method that achieves finite-time guarantees in obtaining $(\delta, \epsilon)$-stationarity for an $L$-Lipschitz function $f : \mathbb{R}^d \to \mathbb{R}$. The main construction we use is the Goldstein subdifferential [20].

**Definition 1** (Goldstein subdifferential). *Consider a locally Lipschitz function $f : \mathbb{R}^d \to \mathbb{R}$, a point $x \in \mathbb{R}^d$, and a parameter $\delta > 0$. The* Goldstein subdifferential *of $f$ at $x$ is the set*

$$\partial_\delta f(x) \stackrel{\text{def}}{=} \text{conv} \Big( \bigcup_{y \in \mathbb{B}_\delta(x)} \partial f(y) \Big).$$

*A point $x$ is called $(\delta, \epsilon)$-stationary if $dist(0, \partial_\delta f(x)) \le \epsilon$.*

Thus, the Goldstein subdifferential of $f$ at $x$ is the convex hull of all Clarke subgradients at points in a $\delta$-ball around $x$. Famously, [20] showed that one can significantly decrease the value of $f$ by taking a step in the direction of the minimal norm element of $\partial_\delta f(x)$. Throughout the rest of the section, we fix $\delta \in (0, 1)$ and use the notation

$$\hat{g} \stackrel{\text{def}}{=} g/\|g\|_2 \text{ for any nonzero vector } g \in \mathbb{R}^d. \tag{3}$$

**Theorem 1** ([20]). *Fix a point $x$, and let $g$ be a minimal norm element of $\partial_\delta f(x)$. Then as long as $g \neq 0$, we have $f(x - \delta\hat{g}) \le f(x) - \delta\|g\|_2$.*

Theorem 1 immediately motivates the following conceptual descent algorithm:

$$x_{t+1} = x_t - \delta\hat{g}_t, \text{ where } g_t \in \underset{g \in \partial_\delta f(x)}{\text{argmin}} \|g\|_2. \tag{4}$$

In particular, Theorem 1 guarantees that, defining $\Delta \stackrel{\text{def}}{=} f(x_0) - \min f$, the *approximate stationarity condition*

$$\min_{t=1,\dots,T} \|g_t\|_2 \le \epsilon \text{ holds after } T = \mathcal{O}\left(\frac{\Delta}{\delta\epsilon}\right) \text{ iterations of (4)}.$$

Evaluating the minimal norm element of $\partial_\delta f(x)$ is impossible in general, and therefore the descent method described in (4) cannot be applied directly. Nonetheless it serves as a guiding principle for implementable algorithms. Notably, the gradient sampling algorithm [29] in each iteration forms polyhedral approximations $K_t$ of $\partial_\delta f(x_t)$ by sampling gradients in the ball $\mathbb{B}_\delta(x)$ and computes search directions $g_t \in \text{argmin}_{g \in K_t} \|g\|_2$. These gradient sampling algorithms, however, have only asymptotic convergence guarantees [26].

The recent paper [1] remarkably shows that for any $x \in \mathbb{R}^d$ one *can* find an *approximate* minimal norm element of $\partial_\delta f(x)$ using a number of subgradient computations that is independent of the dimension. The idea of their procedure is as follows. Suppose that we have a trial vector $g \in \partial_\delta f(x)$ (not necessarily a minimal norm element) satisfying

$$f(x - \delta\hat{g}) \ge f(x) - \frac{\delta}{2}\|g\|_2. \tag{5}$$

That is, the decrease in function value is not as large as guaranteed by Theorem 1 for the true minimal norm subgradient. One would like to now find a vector $u \in \partial_\delta f(x)$ so that the norm of some convex combination $(1 - \lambda)g + \lambda u$ is smaller than that of $g$. A short computation shows that this is sure to be the case for all small $\lambda > 0$ as long as $\langle u, g \rangle \le \|g\|_2^2$. The task therefore reduces to:

$$\text{find some } u \in \partial_\delta f(x) \quad \text{satisfying} \quad \langle u, g \rangle \le \|g\|_2^2.$$

The ingenious idea of [1] is a randomized procedure for establishing exactly that in expectation. Namely, suppose for the moment that $f$ happens to be differentiable along the segment $[x, x - \delta\hat{g}]$; we will revisit this assumption shortly. Then the fundamental theorem of calculus, in conjunction with (5), yields

$$\frac{1}{2}\|g\|_2 \ge \frac{f(x) - f(x - \delta\hat{g})}{\delta} = \frac{1}{\delta}\int_0^\delta \langle \nabla f(x - \tau\hat{g}), \hat{g} \rangle \, d\tau. \tag{6}$$

Consequently, a point $y$ chosen uniformly at random in the segment $[x, x - \delta \hat{g}]$ satisfies

$$\mathbb{E}\langle \nabla f(y), g \rangle \leq \frac{1}{2} \|g\|_2^2. \tag{7}$$

Therefore the vector $u = \nabla f(y)$ can act as the subgradient we seek. Indeed, the following lemma shows that, in expectation, the minimal norm element of $[g, u]$ is significantly shorter than $g$. The proof is extracted from that of [1, Theorem 8].

**Lemma 2** ([1]). *Fix a vector $g \in \mathbb{R}^d$, and let $u \in \mathbb{R}^d$ be a random vector satisfying $\mathbb{E}\langle u, g \rangle < \frac{1}{2}\|g\|_2^2$. Suppose moreover that the inequality $\|g\|_2, \|u\|_2 \leq L$ holds for some $L < \infty$. Then the minimal-norm vector $z$ in the segment $[g, u]$ satisfies:*

$$\mathbb{E}\|z\|_2^2 \leq \|g\|_2^2 - \frac{\|g\|_2^4}{16L^2}.$$

*Proof.* Applying $\mathbb{E}\langle u, g \rangle \leq \frac{1}{2}\|g\|_2^2$ and $\|g\|_2, \|u\|_2 \leq L$, we have, for any $\lambda \in (0, 1)$,

$$\mathbb{E}\|z\|_2^2 \leq \mathbb{E}\|g + \lambda(u - g)\|_2^2 = \|g\|_2^2 + 2\lambda\mathbb{E}\langle g, u - g \rangle + \lambda^2 \mathbb{E}\|u - g\|_2^2$$
$$\leq \|g\|_2^2 - \lambda\|g\|_2^2 + 4\lambda^2 L^2.$$

Plugging in the value $\lambda = \frac{\|g\|_2^2}{8L^2} \in (0, 1)$ minimizes the right hand side and completes the proof. $\qquad\square$

The last technical difficulty to overcome is the requirement that $f$ be differentiable along the line segment $[g, u]$. This assumption is crucially used to obtain (6) and (7). To cope with this problem, [1] introduce extra assumptions on the function $f$ to be minimized and assume a nonstandard oracle access to subgradients.

We show, using Lemma 3, that no extra assumptions are needed if one slightly perturbs $g$.

**Lemma 3.** *Let $f\colon \mathbb{R}^d \to \mathbb{R}$ be a Lipschitz function, and fix a point $x \in \mathbb{R}^d$. Then there exists a set $\mathcal{D} \subset \mathbb{R}^d$ of full Lebesgue measure such that for every $y \in \mathcal{D}$, the line spanned by $x$ and $y$ intersects $\mathrm{dom}(\nabla f)$ in a full Lebesgue measure set in $\mathbb{R}$. Then, for every $y \in \mathcal{D}$ and all $\tau \in \mathbb{R}$, we have*

$$f(x + \tau(y - x)) - f(x) = \int_0^\tau \langle \nabla f(x + s(y - x)), y - x \rangle \, ds.$$

*Proof.* Without loss of generality, we may assume $x = 0$ and $f(x) = 0$. Rademacher's theorem guarantees that $\mathrm{dom}(\nabla f)$ has full Lebesgue measure in $\mathbb{R}^d$. Fubini's theorem then directly implies that there exists a set $\mathcal{Q} \subset \mathbb{S}^{d-1}$ of full Lebesgue measure within the sphere $\mathbb{S}^{d-1}$ such that for every $y \in \mathcal{Q}$, the intersection $\mathbb{R}_+\{y\} \cap (\mathrm{dom}(\nabla f))^c$ is Lebesgue null in $\mathbb{R}$. It follows immediately that the set $\mathcal{D} = \{\tau y : \tau > 0, y \in Q\}$ has full Lebesgue measure in $\mathbb{R}^d$. Fix now a point $y \in \mathcal{D}$ and any $\tau \in \mathbb{R}_+$. Since $f$ is Lipschitz, it is absolutely continuous on any line segment and therefore

$$f(x + \tau(y - x)) - f(x) = \int_0^\tau f'(x + s(y - x), y - x) \, ds = \int_0^\tau \langle \nabla f(x + s(y - x)), y - x \rangle \, ds.$$

The proof is complete. $\qquad\square$

We now have all the ingredients to present a modification of the algorithm from [1], which, under a standard first-order oracle model, either significantly decreases the objective value or finds an approximate minimal norm element of $\partial_\delta f$.

The following theorem establishes the efficiency of Algorithm 1, and its proof is similar to that of [1, Lemma 13]. For completeness, we include the full proof in Appendix A.

**Theorem 4.** *Let $\{g_k\}$ be generated by $\mathtt{MinNorm}(x, \delta, \epsilon)$. Fix an index $k \geq 0$, and define the stopping time $\tau \stackrel{\text{def}}{=} \inf \{k\colon f(x - \delta \hat{g}_k) < f(x) - \delta\|g_k\|_2/4 \text{ or } \|g_k\|_2 \leq \epsilon\}$. Then, we have*

$$\mathbb{E}\left[\|g_k\|_2^2 \mathbb{1}_{\tau > k}\right] \leq \frac{16L^2}{16 + k}.$$

An immediate consequence of Theorem 4 is that $\mathtt{MinNorm}(x, \delta, \epsilon)$ terminates with high-probability.

---

**Algorithm 1** MinNorm$(x, \delta, \epsilon)$

---

1: **Input.** $x$, $\delta > 0$, and $\epsilon > 0$.
2: Let $k = 0$, $g_0 = \nabla f(\zeta_0)$ where $\zeta_0 \sim \mathbb{B}_\delta(x)$.
3: **while** $\|g_k\|_2 > \epsilon$ and $\frac{\delta}{4}\|g_k\|_2 \geq f(x) - f(x - \delta\hat{g}_k)$ **do**
4: $\quad$ Choose any $r$ satisfying $0 < r < \|g_k\|_2 \cdot \sqrt{1 - (1 - \frac{\|g_k\|_2^2}{128L^2})^2}$.
5: $\quad$ Sample $\zeta_k$ uniformly from $\mathbb{B}_r(g_k)$.
6: $\quad$ Choose $y_k$ uniformly at random from the segment $[x, x - \delta\widehat{\zeta_k}]$.
7: $\quad$ $g_{k+1} = \operatorname{argmin}_{z \in [g_k, \nabla f(y_k)]} \|z\|_2$.
8: $\quad$ $k = k + 1$.
9: **end while**
10: Return $g_k$.

---

**Corollary 5.** MinNorm$(x, \delta, \epsilon)$ *terminates in at most* $\left\lceil \frac{64L^2}{\epsilon^2} \right\rceil \cdot \lceil 2\log(1/\gamma) \rceil$ *iterations with probability at least* $1 - \gamma$*, where we define the stopping time* $\tau \stackrel{\text{def}}{=}$ $\inf\{k\colon f(x - \delta\hat{g}_k) < f(x) - \delta\|g_k\|_2/4 \text{ or } \|g_k\|_2 \leq \epsilon\}$.

Combining Algorithm 1 with (4) yields Algorithm 2, with convergence guarantees summarized in Theorem 6, whose proof is identical to that of [1, Theorem 8].

---

**Algorithm 2** Interpolated Normalized Gradient Descent (INGD$(x_0, T)$)

---

Initial $x_0$, counter $T$
**for** $t = 0, \ldots, T - 1$ **do**
$\quad$ $g = $ MinNorm$(x_t)$ $\qquad\qquad\qquad\qquad\qquad\qquad$ ▷ Computational complexity $\widetilde{\mathcal{O}}(L^2/\epsilon^2)$
$\quad$ Set $x_{t+1} = x_t - \delta\hat{g}$ $\qquad\qquad\qquad\qquad\qquad\qquad\qquad$ ▷ We define $\hat{g}$ in (3)
**end for**
Return $x_T$

---

**Theorem 6.** *Fix an initial point* $x_0 \in \mathbb{R}^d$*, and define* $\Delta = f(x_0) - \inf_x f(x)$*. Set the number of iterations* $T = \frac{4\Delta}{\delta\epsilon}$*. Then, with probability* $1 - \gamma$*, the point* $x_T = $ INGD$(x_0, T)$ *satisfies* $\operatorname{dist}(0, \partial_\delta f(x_T)) \leq \epsilon$ *in a total of at most*

$$\left\lceil \frac{4\Delta}{\delta\epsilon} \right\rceil \cdot \left\lceil \frac{64L^2}{\epsilon^2} \right\rceil \cdot \left\lceil 2\log\left(\frac{4\Delta}{\gamma\delta\epsilon}\right) \right\rceil \qquad \textit{function-value and gradient evaluations.}$$

In summary, the complexity of finding a point $x$ satisfying $\operatorname{dist}(0, \partial_\delta f(x)) \leq \epsilon$ is at most $\mathcal{O}\left(\frac{\Delta L^2}{\delta\epsilon^3} \log\left(\frac{4\Delta}{\gamma\delta\epsilon}\right)\right)$ with probability $1 - \gamma$. Using the identity $\partial f(x) = \limsup_{\delta \to 0} \partial_\delta f(x)$, this result also provides a strategy for finding a Clarke stationary point, albeit with no complexity guarantee. It is thus natural to ask whether one may efficiently find some point $x$ for which there exists $y \in \mathbb{B}_\delta(x)$ satisfying $\operatorname{dist}(0, \partial f(y)) \leq \epsilon$. This is exactly the guarantee of subgradient methods on weakly convex functions in [6]. [30] shows that for general Lipschitz functions, the number of subgradient computations required to achieve this goal by any algorithm scales with the dimension of the ambient space. Finally, we mention that the perturbation technique similarly applies to the stochastic algorithm of [1, Algorithm 2], yielding a method that matches their complexity estimate.

## 3 Faster INGD in low dimensions

In this section, we describe our modification of Algorithm 1 ("INGD") for obtaining improved runtimes in the low-dimensional setting. Our modified algorithm hinges on computations similar to (5), (6), and (7) except for the constants involved, and hence we explicitly state this setup. Given a vector $g \in \partial_\delta f(x)$, we say it satisfies the *descent condition* at $x$ if

$$f(x - \delta\hat{g}) \leq f(x) - \frac{\delta\epsilon}{3}. \tag{8}$$

Recall that Lemma 3 shows that for almost all $g$, we have

$$f(x) - f(x - \delta\hat{g}) = \int_0^1 \langle \nabla f(x - t\delta\hat{g}), \hat{g} \rangle \, dt = \delta \cdot \mathbb{E}_{z \sim \text{Unif}[x - \delta\hat{g}, x]} \langle \nabla f(z), \hat{g} \rangle.$$

Hence, when $g$ does *not* satisfy the descent condition (8), we can output a random vector $u \in \partial_\delta f(x)$ such that

$$\mathbb{E}\langle u, g \rangle \leq \frac{\epsilon}{3} \|g\|_2. \tag{9}$$

Then, an arbitrary vector $g$ either satisfies (8) or can be used to output a random vector $u$ satisfying (9). As described in Corollary 5, Algorithm 1 achieves this goal in $\widetilde{\mathcal{O}}(L^2/\epsilon^2)$ iterations.

In this section, we improve upon this oracle complexity by applying cutting plane methods (which we review shortly) to design Algorithm 3, which finds a better descent direction in $\widetilde{\mathcal{O}}(Ld/\epsilon)$ oracle calls for $L$-Lipschitz functions and $\mathcal{O}(d \log(L/\epsilon) \log(\delta\rho/\epsilon))$ oracle calls for $\rho$-weakly convex functions.

**Brief overview of cutting-plane methods.** We first provide a brief relevant overview of cutting-plane methods here and refer the reader to standard textbooks in optimization for a more in-depth exposition. Given a convex function $f$ with its set $\mathcal{S}$ of minimizers, a cutting-plane method (CPM) minimizes $f$ by maintaining a convex search set $\mathcal{E}^{(k)} \supseteq \mathcal{S}$ in the $k^{\text{th}}$ iteration and iteratively shrinking $\mathcal{E}^{(k)}$ guided by the subgradients of $f$ that act as "separation oracles" for the set $\mathcal{S}$. Specifically, this is achieved by noting that for any $x^{(k)}$ chosen from $\mathcal{E}^{(k)}$, if the gradient oracle indicates $\nabla f(x^{(k)}) \neq 0$, (i.e. $x^{(k)} \notin \mathcal{S}$), then the convexity of $f$ guarantees $\mathcal{S} \subseteq \mathcal{H}^{(k)} : \{ y : \langle \nabla f(x^{(k)}), y - x^{(k)} \rangle \leq 0 \}$, and hence $\mathcal{S} \subseteq \mathcal{H}^{(k)} \cap \mathcal{E}^{(k)}$. The algorithm continues by choosing $\mathcal{E}^{(k+1)} \supseteq \mathcal{E}^{(k)} \cap \mathcal{H}^{(k)}$, and different choices of $x^{(k)}$ and $\mathcal{E}^{(k)}$ yield different rates of shrinkage of $\mathcal{E}^{(k)}$ until a point in $\mathcal{S}$ is found.

In light of this description, the minimization of a convex function over a constrained convex set via this cutting-plane method requires, at each iteration, merely a subgradient of the function. Our novel insight is that a lack of function decrease implies we have roughly such a subgradient, which we may then use in a cutting-plane method for computing the minimum norm element of the subdifferential faster in low dimensions (with improved complexity for weakly convex functions).

**Setting the stage for our algorithm.** In Appendix B, we demonstrate how to remove the expectation in (9) and turn the inequality into a high probability statement. For now, we assume the existence of an oracle $\mathscr{O}$ as in Definition 2.

**Definition 2** (Inner Product Oracle). *Given a vector $g \in \partial_\delta f(x)$ that does not satisfy the descent condition (8), the inner product oracle $\mathscr{O}(g)$ outputs a vector $u \in \partial_\delta f(x)$ such that*

$$\langle u, g \rangle \leq \frac{\epsilon}{2} \|g\|_2.$$

We defer the proof of the lemma below to Appendix B.

**Lemma 7.** *Fix $x \in \mathbb{R}^d$ and a unit vector $\hat{g} \in \mathbb{R}^d$ such that $f$ is differentiable almost everywhere on the line segment $[x, y]$, where $y \stackrel{\text{def}}{=} x - \delta\hat{g}$. Suppose that $z \in \mathbb{R}^d$ sampled uniformly from $[x, y]$ satisfies $\mathbb{E}_z \langle \nabla f(z), \hat{g} \rangle \leq \frac{\epsilon}{3}$. Then we can find $\bar{z} \in \mathbb{R}^d$ using at most $O(\frac{L}{\epsilon} \log(1/\gamma))$ gradient evaluations of $f$, such that with probability at least $1 - \gamma$ the estimate $\langle \nabla f(\bar{z}), \hat{g} \rangle \leq \frac{\epsilon}{2}$ holds. Moreover, if $f$ is $\rho$-weakly convex, we can find $\bar{z} \in \mathbb{R}^d$ such that $\langle \nabla f(\bar{z}), \hat{g} \rangle \leq \frac{\epsilon}{2}$ using only $O(\log(\delta\rho/\epsilon))$ function evaluations of $f$.*

Our key insight is that this oracle is almost identical to the gradient oracle of the minimal norm element problem

$$\min_{g \in \partial_\delta f(x)} \|g\|_2.$$

Therefore, we can use it in the cutting plane method to find an approximate minimal norm element of $\partial_\delta f$. When there is no element of $\partial_\delta f$ with norm less than $\epsilon$, our algorithm will instead find a vector that satisfies the descent condition. The main result of this section is the following theorem.

**Theorem 8.** *Let $f : \mathbb{R}^d \to \mathbb{R}$ be an $L$-Lipschitz function. Fix an initial point $x_0 \in \mathbb{R}^d$, and let $\Delta \stackrel{\text{def}}{=} f(x_0) - \inf_x f(x)$. Then, there exists an algorithm that outputs a point $x \in \mathbb{R}^d$ satisfying $\text{dist}(0, \partial_\delta f(x)) \leq \epsilon$ and, with probability at least $1 - \gamma$, uses at most*

$$\mathcal{O}\left( \frac{\Delta L d}{\delta \epsilon^2} \cdot \log(L/\epsilon) \cdot \log(1/\gamma) \right) \qquad \textit{function value/gradient evaluations.}$$

*If $f$ is $\rho$-weakly convex, the analogous statement holds with probability one and with the improved efficiency estimate $\mathcal{O}\left(\frac{\Delta d}{\delta\epsilon}\log(L/\epsilon)\cdot\log(\delta\rho/\epsilon)\right)$ of function value/gradient evaluations.*

## 3.1 Finding a minimal norm element

In this section, we show, via Algorithm 3, how to find an approximate minimal norm element of $\partial_\delta f(x)$. Instead of directly working with the minimal norm problem, we note that, by Cauchy-Schwarz inequality and the Minimax Theorem, for any closed convex set $Q$, we have

$$\min_{g\in Q}\|g\|_2 = \min_{g\in Q}\left[\max_{\|v\|_2\leq 1}\langle g,v\rangle\right] = \max_{\|v\|_2\leq 1}\left[\min_{g\in Q}\langle g,v\rangle\right] = \max_{\|v\|_2\leq 1}\phi_Q(v),\qquad(10)$$

where $\phi_Q(v)\overset{\text{def}}{=}\min_{g\in Q}\langle g,v\rangle$, and Lemma 9 formally connects the problem of finding the minimal norm element with that of maximizing $\phi_Q$. The key observation in this section (Lemma 10) is that the inner product oracle $\mathcal{O}$ is a separation oracle for the (dual) problem $\max_{\|v\|_2\leq 1}\phi_Q(v)$ with $Q=\partial_\delta f(x)$ and hence can be used in cutting plane methods.

**Lemma 9.** *Let $Q\subset\mathbb{R}^d$ be a closed convex set that does not contain the origin. Let $g_Q^*$ be a minimizer of $\min_{g\in Q}\|g\|_2$. Then, the vector $v_Q^*=g_Q^*/\|g_Q^*\|_2$ satisfies*

$$\langle v_Q^*,g\rangle\geq\|g_Q^*\|_2\qquad\text{for all }g\in Q.$$

*and $v_Q^*=\arg\max_{\|v\|_2\leq 1}\phi_Q(v)$.*

*Proof.* We omit the subscript $Q$ to simplify notation. Since, by definition, $g^*$ minimizes $\|g\|_2$ over all $g\in Q$, we have
$$\langle g^*,g\rangle\geq\|g^*\|_2^2\text{ for all }g\in Q,$$
and the inequality is tight for $g=g^*$. Using this fact and $\phi(v^*)=\min_{g\in Q}\langle g,\frac{g^*}{\|g^*\|_2}\rangle$ gives

$$\phi(v^*)=\|g^*\|_2=\min_{g\in Q}\|g\|_2=\min_{g\in Q}\max_{v:\|v\|_2\leq 1}\langle g,v\rangle=\max_{\|v\|_2\leq 1}\min_{g\in Q}\langle g,v\rangle=\max_{v:\|v\|_2\leq 1}\phi(v),$$

where we used Sion's minimax theorem in the second to last step. This completes the proof. $\qquad\square$

Using this lemma, we can show that $\mathcal{O}$ is a separation oracle.

**Lemma 10.** *Consider a vector $g\in\partial f_\delta(x)$ that does not satisfy the descent condition (8), and let the output of querying the oracle at $g$ be $u\in\mathcal{O}(g)$. Suppose that $\text{dist}(0,\partial_\delta f(x))\geq\frac{\epsilon}{2}$. Let $g^*$ be the minimal-norm element of $\partial_\delta f(x)$. Then the normalized vector $v^*\overset{\text{def}}{=}g^*/\|g^*\|_2$ satisfies the inclusion:*
$$v^*\in\left\{w\in\mathbb{R}^d:\langle u,\hat{g}-w\rangle\leq 0\right\}.$$

*Proof.* Set $Q=\partial_\delta f(x)$. By using $\langle u,\hat{g}\rangle\leq\frac{\epsilon}{2}$ (the guarantee of $\mathcal{O}$ per Definition 2) and $\langle u,v^*\rangle\geq\|g^*\|_2$ (from Lemma 9), we have $\langle u,\hat{g}-v^*\rangle=\langle u,\hat{g}\rangle-\langle u,v^*\rangle\leq\frac{\epsilon}{2}-\|g^*\|_2\leq 0$. $\qquad\square$

Thus Lemma 10 states that if $x$ is not a $(\delta,\frac{\epsilon}{2})$-stationary point of $f$, then the oracle $\mathcal{O}$ produces a halfspace $\mathcal{H}_v$ that separates $\hat{g}$ from $v^*$. Since $\mathcal{O}$ is a separation oracle, we can combine it with any cutting plane method to find $v^*$. For concreteness, we use the center of gravity method and display our algorithm in Algorithm 3. We note that $\Omega_k$ is an intersection of a ball and some half spaces, hence we can compute its center of gravity in polynomial time by taking an average of the empirical samples from this convex set. While we use a simple cutting-plane method, any algorithm in this class may be used; our focus is merely on minimizing the oracle complexity. Further note that in our algorithm, we use a point $\zeta_k$ close to the true center of gravity of $\Omega_k$, and therefore, we invoke a result about the *perturbed* center of gravity method.

**Theorem 11** (Theorem 3 of [31]; see also [32]). *Let $K$ be a convex set with center of gravity $\mu$ and covariance matrix $A$. For any halfspace $H$ that contains some point $x$ with $\|x-\mu\|_{A^{-1}}\leq t$, we have*

$$\text{vol}(K\cap H)\leq(1-1/e+t)\text{vol}(K).$$

**Algorithm 3** MinNormCG($x$)

---

1: **Initialize** center point $x$.
2: Set $k = 0$, the search region $\Omega_0 = \mathbb{B}_2(0)$, the set of gradients $Q_0 = \{\nabla f(x)\}$, and $r$ satisfying $0 < r < \epsilon/(32dL)$
3: **while** $\min_{g \in Q_k} \|g\|_2 > \epsilon$ **do**
4:     Let $v_k$ be the center of gravity of $\Omega_k$.
5:     **if** $v_k$ satisfies the descent condition (8) at $x$ **then**
6:         Return $v_k$
7:     **end if**
8:     Sample $\zeta_k$ uniformly from $\mathbb{B}_r(v_k)$
9:     $u_k \leftarrow \mathcal{O}(\zeta_k)$
10:    $\Omega_{k+1} = \Omega_k \cap \{w : \langle u_k, \zeta_k - w \rangle \leq 0\}$.
11:    $Q_{k+1} = \text{conv}(Q_k \cup \{u_k\})$
12:    $k = k + 1$
13: **end while**
14: Return $\arg\min_{g \in Q_k} \|g\|_2$.

---

**Theorem 12** (Theorem 4.1 of [33]). *Let $K$ be a convex set in $\mathbb{R}^d$ with center of gravity $\mu$ and covariance matrix $A$. Then,*

$$K \subset \left\{ x : \|x - \mu\|_{A^{-1}} \leq \sqrt{d(d+2)} \right\}.$$

We now have all the tools to show correctness and iteration complexity of Algorithm 3.

**Theorem 13.** *Let $f : \mathbb{R}^d \to \mathbb{R}$ be an L-Lipschitz function. Then Algorithm 3 returns a vector $v \in \partial_\delta f(x)$ that either satisfies the descent condition (8) at $x$ or satisfies $\|v\|_2 \leq \epsilon$ in*

$$\lceil 8d \log(8L/\epsilon)) \rceil \text{ calls to } \mathcal{O}.$$

*Proof.* By the description of Algorithm 3, either it returns a vector $v$ satisfying the descent condition or returns $g \in \partial_\delta f(x)$ with $\|g\|_2 \leq \epsilon$. We now obtain the algorithm's claimed iteration complexity.

Consider an iteration $k$ such that $\Omega_k$ *does* contain a ball of radius $\frac{\epsilon}{4L}$. Let $A_k$ be the covariance matrix of convex set $\Omega_k$. By Theorem 12, we have

$$A_k \succeq \left( \frac{\epsilon}{8dL} \right)^2 I.$$

Applying this result to the observation that in Algorithm 3 $\zeta_k$ is sampled uniformly from $\mathbb{B}_r(v_k)$ gives

$$\|v_k - \zeta_k\|_{A_k^{-1}} \leq r \cdot \frac{8dL}{\epsilon} \leq \frac{1}{4}. \tag{11}$$

Recall from Algorithm 3 and the preceding notation that $\Omega_k$ has center of gravity $v_k$ and covariance matrix $A_k$. Further, the halfspace $\{w : \langle u_k, \zeta_k - w \rangle \leq 0\}$ in Algorithm 3 contains the point $\zeta_k$ satisfying (11). Given these statements, since Algorithm 3 sets $\Omega_{k+1} = \Omega_k \cap \{w : \langle u_k, \zeta_k - w \rangle\}$, we may invoke Theorem 11 to obtain

$$\text{vol}(\Omega_k) \leq (1 - 1/e + 1/4)^k \text{vol}(\mathbb{B}_2(0)) \leq (1 - 1/10)^k \text{vol}(\mathbb{B}_2(0)). \tag{12}$$

We claim that Algorithm 3 takes at most $T + 1$ steps where $T = d \log_{(1-\frac{1}{10})}(\epsilon/(8L))$. For the sake of contradiction, suppose that this statement is false. Then, applying (12) with $k = T + 1$ gives

$$\text{vol}(\Omega_{T+1}) \leq \left( \frac{\epsilon}{4L} \right)^d \text{vol}(\mathbb{B}_1(0)). \tag{13}$$

On the other hand, Algorithm 3 generates points $u_i = \mathcal{O}(\zeta_i)$ in the $i$-th call to $\mathcal{O}$ and the set $Q_i = \text{conv}\{u_1, u_2, \cdots, u_i\}$. Since we assume that the algorithm takes more than $T + 1$ steps, we have $\min_{g \in Q_{T+1}} \|g\|_2 \geq \epsilon$. Using this and $u_i \in Q_{T+1}$, Lemma 10 lets us conclude that $v^*_{Q_{T+1}} \in \{w \in \mathbb{R}^d : \langle u_i, \zeta_i - w \rangle \leq 0\}$ for all $i \in [T+1]$. Since $\Omega_{T+1}$ is the intersection of the unit ball and these halfspaces, we have

$$v^*_{Q_{T+1}} \in \Omega_{T+1}.$$

Per (13), $\Omega_{T+1}$ does not contain a ball of radius $\frac{\epsilon}{4L}$, and therefore we may conclude that

$$\text{there exists a point } \widetilde{v} \in \mathbb{B}_{\frac{\epsilon}{2L}}(v^*_{Q_{T+1}}) \text{ such that } \widetilde{v} \notin \Omega_{T+1}.$$

Since $\widetilde{v} \in \mathbb{B}_2(0)$, the fact $\widetilde{v} \notin \Omega_{T+1}$ must be true due to one of the halfspaces generated in Algorithm 3. In other words, there must exist some $i \in [T+1]$ with

$$\langle u_i, \zeta_i - \widetilde{v} \rangle > 0.$$

By the guarantee of $\mathcal{O}$, we have $\langle u_i, \zeta_i \rangle \leq \frac{\epsilon}{2}$, and hence

$$\langle u_i, \widetilde{v} \rangle = \langle u_i, v_i \rangle - \langle u, v_i - \widetilde{v} \rangle < \frac{\epsilon}{2}. \tag{14}$$

By applying $\widetilde{v} \in \mathbb{B}_{\frac{\epsilon}{2L}}(v^*_{Q_{T+1}})$, $u_i \in \partial_\delta f(x)$, $L$-Lipschitzness of $f$, and Lemma 9, we have

$$\langle u_i, \widetilde{v} \rangle \geq \langle u_i, v^*_{Q_{T+1}} \rangle - \frac{\epsilon}{2L} \|u_i\|_2 \geq \langle u_i, v^*_{Q_{T+1}} \rangle - \frac{\epsilon}{2} \geq \|g^*_{Q_{T+1}}\|_2 - \frac{\epsilon}{2}. \tag{15}$$

Combining (14) and (15) yields that $\min_{g \in Q_{T+1}} \|g\|_2 = \|g^*_{Q_{T+1}}\|_2 < \epsilon$. This contradicts the assumption that the algorithm takes more than $T+1$ steps and concludes the proof. $\qquad\square$

Now, we are ready to prove the main theorem.

*Proof of Theorem 8.* We note that the outer loop in Algorithm 2 runs at most $\mathcal{O}(\frac{\Delta}{\delta\epsilon})$ times because we decrease the objective by $\Omega(\delta\epsilon)$ every step. Combining this with Theorem 13 and Lemma 7, we have that with probability $1 - \gamma$, the oracle complexity for $L$-Lipschitz function is

$$\left\lceil \frac{4\Delta}{\delta\epsilon} \right\rceil \cdot \lceil 8d \log(8L/\epsilon) \rceil \cdot \left\lceil \frac{36L}{\epsilon} \right\rceil \cdot \left\lceil 2\log\left(\frac{4\Delta}{\gamma\delta\epsilon}\right) \right\rceil = \mathcal{O}\left(\frac{\Delta L d}{\delta\epsilon^2} \cdot \log(L/\epsilon) \cdot \log(1/\gamma)\right)$$

and for $L$-Lipschitz and $\rho$-weakly convex function is $\mathcal{O}\left(\frac{\Delta d}{\delta\epsilon} \log(L/\epsilon) \cdot \log(\delta\rho/\epsilon)\right)$.

$\qquad\square$

## Acknowledgements

Research of D. Davis was supported by an Alfred P. Sloan research fellowship and NSF DMS award 2047637. Research of D. Drusvyatskiy was supported by NSF DMS-1651851 and CCF-2023166 awards. Research of YL was supported by NSF awards CCF-1749609, DMS-1839116, DMS-2023166, CCF-2105772, a Microsoft Research Faculty Fellowship, Sloan Research Fellowship, and Packard Fellowship. Research of GY was supported by an MIT Presidential Fellowship. Part of this work was done while GY was a student at the University of Washington.

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
