## A  Missing proofs from Section 2

**Theorem 4.** *Let $\{g_k\}$ be generated by* MinNorm$(x, \delta, \epsilon)$. *Fix an index $k \geq 0$, and define the stopping time $\tau \stackrel{\text{def}}{=} \inf\{k\colon f(x - \delta\hat{g}_k) < f(x) - \delta\|g_k\|_2/4 \text{ or } \|g_k\|_2 \leq \epsilon\}$. Then, we have*

$$\mathbb{E}\left[\|g_k\|_2^2 1_{\tau > k}\right] \leq \frac{16L^2}{16 + k}.$$

*Proof.* Fix an index $k$, and let $\mathbb{E}_k[\cdot]$ denote the conditional expectation on $g_k$. Suppose we are in the event $\{\tau > k\}$. Taking into account the Lipschitz continuity of $f$ and Lemma 3, we deduce that almost surely, conditioned on $g_k$, the following estimate holds:

$$\begin{aligned}
\frac{1}{4}\|g_k\|_2 \geq \frac{f(x) - f(x - \delta\hat{g}_k)}{\delta} &\geq \frac{f(x) - f(x - \delta \cdot \hat{\zeta}_k)}{\delta} - L\|\hat{g}_k - \hat{\zeta}_k\|_2 \\
&= \frac{1}{\delta}\int_0^\delta \langle \nabla f(x - s\hat{\zeta}_k), \hat{\zeta}_k\rangle \, ds - L\|\hat{g}_k - \hat{\zeta}_k\|_2 \\
&\geq \frac{1}{\delta}\int_0^\delta \langle \nabla f(x - s\hat{\zeta}_k), \hat{g}_k\rangle \, ds - 2L\|\hat{g}_k - \hat{\zeta}_k\|_2 \\
&= \mathbb{E}_k\langle \nabla f(y_k), \hat{g}_k\rangle - 2L\|\hat{g}_k - \hat{\zeta}_k\|_2.
\end{aligned}$$

Rearranging yields $\mathbb{E}_k\langle \nabla f(y_k), \hat{g}_k\rangle \leq \frac{1}{4}\|g_k\|_2 + 2L\|\hat{g}_k - \hat{\zeta}_k\|$. Simple algebra shows $\|\hat{g}_k - \hat{\zeta}_k\|_2^2 \leq 2(1 - \sqrt{1 - r^2/\|g_k\|_2^2}) \leq \frac{\|g_k\|_2^2}{64L^2}$. Therefore, we infer that $\mathbb{E}_k\langle \nabla f(y_k), \hat{g}_k\rangle < \frac{1}{2}\|g_k\|_2$. Lemma 2 then guarantees that

$$\mathbb{E}_k[\|g_{k+1}\|_2^2 1_{\tau > k}] \leq \left(\|g_k\|_2^2 - \frac{\|g_k\|_2^4}{16L^2}\right)1_{\tau > k}.$$

Define $b_k := \|g_k\|_2^2 1_{\tau > k}$ for all $k \geq 0$. Then the tower rule for expectations yields

$$\mathbb{E}b_{k+1} \leq \mathbb{E}[\|g_{k+1}\|_2^2 1_{\tau > k}] \leq \mathbb{E}\left[\left(1 - \frac{b_k}{16L^2}\right) b_k\right] \leq \left(1 - \frac{\mathbb{E}b_k}{16L^2}\right) \mathbb{E}b_k,$$

by Jensen's inequality applied to the concave function $t \mapsto (1 - t/16L^2)t$. Setting $a_k = \mathbb{E}b_k/L^2$, this inequality becomes $a_{k+1} \leq a_k - a_k^2/16$, which, upon rearranging, yields $\frac{1}{a_{k+1}} \geq \frac{1}{a_k(1-a_k/16)} \geq \frac{1}{a_k} + \frac{1}{16}$. Iterating the recursion and taking into account $a_0 \leq 1$ completes the proof. $\qquad \square$

**Corollary 14.** $\mathtt{MinNorm}(x, \delta, \epsilon)$ *terminates in at most* $\left\lceil \frac{64L^2}{\varepsilon^2} \right\rceil \cdot \lceil 2\log(1/\gamma) \rceil$ *iterations with probability at least* $1 - \gamma$, *where we define the stopping time* $\tau \stackrel{\text{def}}{=} \inf \{k : f(x - \delta\hat{g}_k) < f(x) - \delta\|g_k\|_2/4 \text{ or } \|g_k\|_2 \leq \epsilon\}$.

*Proof.* Notice that when $k \geq \frac{64L^2}{\varepsilon^2}$, we have, by Theorem 4, that

$$\Pr(\tau > k) \leq \Pr(\|g_k\|_2 1_{\tau > k} \geq \epsilon) \leq \frac{16L^2}{(16+k)\varepsilon^2} \leq \frac{1}{4}.$$

Similarly, for all $i \in \mathbb{N}$, we have $\Pr(\tau > ik \mid \tau > (i-1)k) \leq 1/4$. Therefore,

$$\Pr(\tau > ik) = \Pr(\tau > ik \mid \tau > (i-1)k)\Pr(\tau > (i-1)k) \leq \frac{1}{4}\Pr(\tau > (i-1)k) \leq \frac{1}{4^i}.$$

Consequently, we have $\Pr(\tau > ik) \leq \frac{1}{4^i} \leq \gamma$ whenever $i \geq \log(1/\gamma)/\log(4)$, as desired. $\qquad \square$

# B  Implementation of the oracles: proof of Lemma 7

In this section, we show how to convert (9) into a deterministic guarantee.

**Lemma 15.** *Fix a unit vector* $\hat{g} \in \mathbb{R}^d$ *and let* $z \in \mathbb{R}^d$ *be a random vector satisfying* $\mathbb{E}\langle \nabla f(z), \hat{g} \rangle \leq \frac{\epsilon}{3}$. *Let* $z_1, \ldots, z_k$ *be i.i.d realizations of* $z$ *with* $k = \left\lceil \frac{36L}{\epsilon} \right\rceil \cdot \left\lceil \frac{\log(1/\gamma)}{\log(4)} \right\rceil$. *Then with probability at least* $1 - \gamma$, *one of the samples* $z_i$ *satisfies* $\langle \nabla f(z_i), \hat{g} \rangle \leq \frac{\epsilon}{2}$.

*Proof.* Define the random variable $Y \stackrel{\text{def}}{=} \langle \nabla f(z), \hat{g} \rangle$, and use $p \stackrel{\text{def}}{=} \Pr[Y \leq \frac{\epsilon}{2}]$. We note that

$$\mathbb{E}[Y] = p \cdot \mathbb{E}[Y \mid Y \leq \frac{\epsilon}{2}] + (1 - p) \cdot \mathbb{E}[Y \mid Y > \frac{\epsilon}{2}].$$

Rearranging the terms and using $\mathbb{E}[Y] \leq \epsilon/3$ gives

$$p \cdot \left(\mathbb{E}[Y \mid Y > \frac{\epsilon}{2}] - \mathbb{E}[Y \mid Y \leq \frac{\epsilon}{2}]\right) \geq \frac{\epsilon}{6}.$$

Finally, taking into account that $f$ is $L$-Lipschitz, we deduce $|Y| \leq L$, which further implies $p \geq \frac{\epsilon}{12L}$. The results follows immediately. $\qquad \square$

**Lemma 16.** *Let* $f: \mathbb{R}^d \to \mathbb{R}$ *be an* $L$-*Lipschitz continuous and* $\rho$-*weakly convex function. Fix a point* $x$ *and a unit vector* $\hat{g} \in \mathbb{R}^d$ *such that* $f$ *is differentiable almost everywhere on the line segment* $[x, y]$, *where* $y \stackrel{\text{def}}{=} x - \delta\hat{g}$. *Suppose that a random vector* $z$ *sampled uniformly from* $[x, y]$ *satisfies* $\mathbb{E}_z\langle \nabla f(z), \hat{g} \rangle \leq \frac{\epsilon}{3}$. *Then, Algorithm 4 finds* $\bar{z} \in \mathbb{R}^d$ *such that* $\langle \nabla f(\bar{z}), \hat{g} \rangle \leq \frac{\epsilon}{2}$ *using* $3\log(12\delta\rho/\epsilon)$ *function evaluations of* $f$.

*Proof.* Define the new function $h : [0, 1] \to \mathbb{R}$ by $h(t) = \langle \nabla f(x + t(y - x)), \hat{g} \rangle$. Clearly, we have

$$\frac{\epsilon}{3} \geq \mathbb{E}[h(t)] = \frac{1}{2}\underbrace{\mathbb{E}[h(t) \mid t \leq 0.5]}_{P_{\leq}} + \frac{1}{2}\underbrace{\mathbb{E}[h(t) \mid t > 0.5]}_{P_{>}}.$$

Therefore $P_{\leq}$ or $P_{>}$ is at most $\epsilon/3$. The fundamental theorem of calculus directly implies $P_{\leq} = \frac{f(x) - f(x - \frac{\delta}{2}\hat{g})}{2\delta}$ and $P_{>} = \frac{f(x - \frac{\delta}{2}\hat{g}) - f(y)}{2\delta}$. Therefore with three function evaluations we may determine

---

**Algorithm 4** Binary Search for $\bar{z}$

---

    **Input.** Line Segment $[x, y = x - \delta\hat{g}]$
    Let $[a, b] = [0, 1]$
    **while** $b - a > \frac{\epsilon}{6\delta\rho}$ **do**
        **if** $f(x - a\delta\hat{g}) - f(x - \frac{a+b}{2}\delta\hat{g}) \leq f(x - \frac{a+b}{2}\delta\hat{g}) - f(x - b\delta\hat{g})$ **then**
            Let $[a, b] \leftarrow [a, \frac{a+b}{2}]$
        **else**
            Let $[a, b] \leftarrow [\frac{a+b}{2}, b]$
        **end if**
    **end while**
    **Return** $x - a\delta\hat{g}$

---

one of the two alternatives. Repeating this procedure $\log(12\delta\rho/\epsilon)$ times, each times shrinking the interval by half, we can find an interval $[a, b] \subset [0, 1]$ such that $b - a \leq \frac{\epsilon}{6\delta\rho}$ and $\mathbb{E}_{t\in[a,b]}h(t) \leq \frac{\epsilon}{3}$. Note that for any $\bar{t} \in [a, b]$, we have $h(\bar{t}) = \mathbb{E}h(t) + (h(\bar{t}) - \mathbb{E}h(t))$, while weak convexity implies

$$h(\bar{t}) - \mathbb{E}h(t) = \frac{1}{\delta}\mathbb{E}_{t\in[a,b]}\langle \nabla f(x + \bar{t}(y - x)) - \nabla f(x + t(y - x)), x - y\rangle$$

$$\leq \mathbb{E}_{t\in[a,b]}\frac{\bar{t} - t}{\delta}\rho\|y - x\|^2 \leq \frac{\epsilon}{6}.$$

We thus conclude $h(\bar{t}) \leq \frac{\epsilon}{3} + \frac{\epsilon}{6} = \frac{\epsilon}{2}$ as claimed. $\qquad\square$