# OpenReview forum: "A gradient sampling method with complexity guarantees for Lipschitz functions in high and low dimensions"
_NeurIPS.cc/2022/Conference — NeurIPS 2022 Accept_

### Official Review · Reviewer_hvUX · 2022-07-11

**Rating:** 7
**Confidence:** 4
**Soundness:** 3 good
**Presentation:** 3 good
**Contribution:** 3 good

**Summary:**

In this paper the authors propose a new algorithm to minimize a Lipschitz function.  The general machinery is based on the [ZLJ 20] algorithm and the main contribution of this paper is to propose a novel implementable oracle for it.


**Questions:**

line 53: extra space between reference and a word.

paragraph after eq (4): either not larger instead of smaller or it should be a strict inequality later. Otherwise, $u=g$ is a counterexample.

Lemma 2: I think the constant can be improved from 16 to 4. It is clear that the given proof is not optimal: we estimate $\|u - g\|$ in a crude way. Instead, we should use $\|u\|^2, \|g\|^2$ separately because for the inner product we have a much nicer bound.  Not important of course, but a few consecutive constants will become smaller.

Lemma 3: We integrate here over $S^{d-1}\times \mathbb{R}_+$ and Fubini's theorem allows us to switch integrals. Why does it give us the existence of the set $Q$? I feel it should be correct but perhaps some argument is missing.

Proof of Lemma 3: different fonts for $Q$.

line 194: Don't we need boundedness of $Q$ for min?

line 204: It is correct, but I spent like 5 min to understand why it is so. Just adding "first-order optimality condition" would save me them.

Section 3.1: I think it would be nicer to add more details. Right now it is written by someone and for someone who already knows all the ingredients: cutting plane algorithms, ellipsoid algorithms, center of gravity, etc. Perhaps with an extra page it would be possible to make that part more readable.

Proof of Th. 4: In the display equation we use Lipschitzness of $f$ twice, why not to mention it?
My simple geometry suggests that the bound for $\| \hat g_k - \hat \zeta\|^2$ contains an extra factor $2$.


**Ethics Review Area:**

["I don’t know"]

**Limitations:**

--

**Strengths And Weaknesses:**

In contrast to [ZLJ 20], the proposed algorithm is easy to implement. This was done without sacrificing a good complexity, which in our case  matches lower bounds. In lower dimensions using the cutting plane algorithm instead of the subgradient method, the authors were able to improve complexity upon [ZLJ 20].

I think it is a nice paper with a clear and well-executed idea and with good explanations that show the motivation for each statement or its proof. However, the last section on low dimensions can be improved: first to make the reading more or less uniform and second to familiarize readers with those concept given there without definitions and intuition. I presume it was the page limit that prevented the authors from doing it in the first place.

---

> ### Author Response · Authors · 2022-07-31
> **Response to Reviewer hvUX**
>
> We thank the reviewer for their appreciation of our paper for its ideas, clarity, implementability, and overall significance. We are also glad to see both our key contributions appreciated.
>
> We are also especially grateful to the reviewer for their very constructive and specific critique on the presentation of our algorithm using the cutting-plane method and corresponding proof details in Section 3. We will add more detail in the proofs in the main body so as to make it read easier and the text below as background material for CPM in the appendix.
>
> ### Background on CPM.
>
> Given a convex function $f$ with its set $\mathcal{S}$ of minimizers, CPM minimizes $f$ by maintaining a convex search set $\mathcal{E}^{(k)}\supseteq \mathcal{S}$ in the $k^{\mathrm{th}}$ iteration and iteratively shrinking $\mathcal{E}^{(k)}$ guided by the subgradients of $f$ that act as “separation oracles” for the set $\mathcal{S}$.
> Specifically, this is achieved by noting that for any $x^{(k)}$ chosen from $\mathcal{E}^{(k)}$, if the gradient oracle indicates $\nabla f(x^{(k)}) \neq 0$, (i.e.\ $x^{(k)}\notin \mathcal{S}$), then the convexity of $f$ guarantees $\mathcal{S} \subseteq \mathcal{H}^{(k)} := \\{y: \nabla f(x^{(k)})^\top(y - x^{(k)}) \leq 0 \\}$, and hence $\mathcal{S} \subseteq \mathcal{H}^{(k)} \cap \mathcal{E}^{(k)}$.
> The algorithm continues by choosing $\mathcal{E}^{(k+1)} \supseteq \mathcal{E}^{(k)} \cap \mathcal{H}^{(k)}$, and different choices of $x^{(k)}$ and $\mathcal{E}^{(k)}$ yield different rates of shrinkage of $\mathcal{E}^{(k)}$ until a point in $\mathcal{S}$ is found.
>
> In light of this description, the minimization of a convex function over a constrained convex set via this cutting-plane method requires, at each iteration, merely a subgradient of the function. Our novel insight is that a lack of function decrease implies we have roughly such a subgradient, which we may then use in a cutting-plane method for computing the minimum norm element of the subdifferential faster in low dimensions (with improved complexity for weakly convex functions).
>
> ### Answers to specific questions
> We now answer the key specific questions posed by the reviewer, and we will incorporate all the fixes on typos caught by the reviewer (thank you again for the careful read and for catching them!).
>
> > paragraph after eq (4): either not larger instead of smaller or it should be a strict inequality later. Otherwise, $u=g$ is a counterexample.
>
> Thank you for catching this, we’ll fix it.
>
> >Lemma 2: I think the constant can be improved from 16 to 4. ... Not important of course, but a few consecutive constants will become smaller.
>
> Thank you for such a careful check of the proofs. We agree that our constants might not be optimal and will incorporate your suggested improvements.
>
> > Lemma 3: ...Fubini's theorem allows us to switch integrals. Why does it give us the existence of the set Q? I feel it should be correct but perhaps some argument is missing.
>
> Let $Z = \mathrm{dom}(\nabla f)^c$ and let $\mu_n$ denote the $n$-dimensional Lebesgue measure. Fubini's Theorem says that $0 = \mu_d(Z) = \int_{\mathcal{S}^{d-1}} \mu_1(Z \cap \mathbb{R}_+\\{y\\})dy.$
>
> Thus $\mu_1(Z \cap \mathbb{R}_+\\{y\\})$ must be zero for almost all $y$ in the sphere.
>
> Thus we may take $Q$ to be any full measure subset of the sphere on which $y \mapsto \mu_1(Z \cap \mathbb{R}_+\\{y\\})$ vanishes.
>
> > line 194: Don't we need boundedness of Q for min?
>
> Could the reviewer please clarify this? There is no min involved in the definition of Q, so it’s unclear to us what’s being referred to
>
> > Proof of Th. 4: In the display equation we use Lipschitzness of f twice, why not to mention it? My simple geometry suggests that the bound for |g^k−ζ^|2 contains an extra factor 2
>
> Thank you for carefully checking our proof; we'll fix this. We note that this doesn't affect the overall correctness but improves some constants.
>
> We hope that these edits of ours, in light of our contributions, elevate the reviewer’s opinion of our submission. Please do let us know if there is something else we can address, and we look forward to further discussions.

---

> > ### Comment · Reviewer_hvUX · 2022-08-07
> > **I am satisfied with authors' response.**
> >
> > I will only give a few remarks.
> >
> > 1. Regarding CPM I meant clarifications not for me, but in general in the paper to make the reading of the paper more or less uniform.
> > 2. By line 194 I meant the equation (9) below, where we take min over $Q$.

---

### Official Review · Reviewer_e5FR · 2022-07-11

**Rating:** 9
**Confidence:** 3
**Soundness:** 4 excellent
**Presentation:** 4 excellent
**Contribution:** 4 excellent

**Summary:**

This paper considers the problem of minimising a Lipschitz function that is potentially non-smooth. Such functions are widely used in machine learning such as neural networks based on the ReLU activation function.
They are known to be differentiable almost everywhere (in Lebesgue measure sense), and the so-called Clarke sub-differential can be used in place of the usual gradient.

The subgradient method is a classical algorithm for minimising such functions, and its behaviour has been well studied for weakly convex functions. Recently, the paper [ZLJ+20] proposed an algorithm for finding stationary points for arbitrary Lipshitz and directionally differentiable functions based on Goldstein's conceptual sub gradient method. Recently, this was also complemented with lower bounds.

Despite its strengths, the algorithm of [ZLJ+20] requires the solution of a so-called auxiliary convex feasibility problem, which may be very difficult when the subgradient of the function is complicated. Hence the algorithm is not easy to implement in practice.

The paper proposes two algorithms that address these concerns by replacing the non-standard auxiliary problem (oracle) of [ZLJ+20] with a standard first-order oracle. This is implementable for any Lipschitz function where the gradients can be computed almost everywhere in terms of Lipschitz measure. The algorithms output an approximate stationary point. The number of gradient/function evaluations for the first algorithm has no dimension dependence, and $O(1/(\epsilon^3 \delta))$ dependence in the approximation accuracy parameters $\epsilon$ and $\delta$. The second algorithm has a better behaviour in low dimensions, with $O(d/(\epsilon^2 \delta))$ dependence, and even better rates for weakly convex functions.

**Questions:**

Do the authors think that this new algorithm could be used for training ReLU-based neural networks?
There have been many efforts recently on coming up with sharp bounds on the Lipschitz constants, but perhaps the computation time for such algorithms could be a limiting factor.

**Limitations:**

The authors have adequately addressed the limitations of their work.

**Strengths And Weaknesses:**

In our opinion, this is a very nice paper that studies a highly practically relevant question on finding stationary points for non-smooth Lipschitz functions.  The contributions of the paper are novel algorithms together with rigorous convergence analysis that obtain state-of-the-art accuracy guarantees for this problem. These algorithms cleverly use randomisation ideas to get around the issues of non-differentiability on a set of Lebesgue measure zero. The paper is very well written and it is a pleasure to read. In our opinion, this is an important contribution to the literature.

This is an important theoretical contribution with a significant amount of effort devoted to the proofs. It is understandable that there is no space in the current NeurIPS format for experimental evaluation, this is something that could be looked at in the future.

---

> ### Author Response · Authors · 2022-07-31
> **Response to Reviewer e5FR**
>
> We thank the reviewer for their careful and detailed review. We agree with their summary of our paper emphasizing both our key contributions, and we are very much encouraged by their appreciation of the paper’s novelty, ideas, mathematical rigour, and presentation, and for their confidence in its significance.
>
> To answer the reviewer’s question
> > Do the authors think that this new algorithm could be used for training ReLU-based neural networks?
>
> we believe that efficient cutting-plane methods could certainly be used in this regard; more broadly, in our opinion, one of the takeaways of this paper is an initiation of the systematic study of the use of cutting-plane methods in the field of non-convex non-smooth optimization.

---

> > ### Comment · Reviewer_e5FR · 2022-08-07
> > **Response to authors**
> >
> > Our question has been answered satisfactorily. We preserve our rating of the paper.

---

### Official Review · Reviewer_4qyS · 2022-07-11

**Rating:** 7
**Confidence:** 3
**Soundness:** 3 good
**Presentation:** 2 fair
**Contribution:** 3 good

**Summary:**

This work studies the problem of finding stationary points of a Lipschitz function. The paper is an improvement to [ZLJ+20], where in the latter the authors proposed an algorithm motivated by Goldstein's subgradient method. The main obstacle of [ZLJ+20] is that the proposed algorithm needs the solutions of a convex feasibility problem (outlined in eq. (2), page 2). Under the assumption that the objective function is Lipschitz and its gradient and function evaluations can be computed almost everywhere, this paper proposed an algorithm that overcomes the aforementioned limitation of [ZLJ+20] and can recover the main result of [ZLJ+20], outlined in Theorem 6, page 6.  Then, the authors claim that the paper can improve the efficiency of the algorithm in low dimensions. The results also provide a further improvement in the case that the objective function is weakly convex (defined in line 19).

**Questions:**

1. Lines 110 - 111: "Famously, [ZLJ+20] showed that one can significantly decrease the value of $f$...". I will appreciate that if the authors address this in [ZLJ+20] and point out where is exactly such a result.

2. In the proof of Lemma 3, what are the definitions of $\mathbb{R}_{+}\{y\}$ and $Q$?

3. In Algorithm 3, line 4: Since $\Omega_k$ may be a complicated set, how one can compute the center of gravity of such a set?

4. In Theorems 11 and 12: What is the definition of the covariance matrix of a convex set?

**Limitations:**

The authors addressed the limitations of their work (lines 98 - 99). I will update this part after the discussion phase and author responses.

**Strengths And Weaknesses:**

Strengths:

1. The main contribution of the paper which is an improvement to [ZLJ+20] is well motivated.
2. The authors provide a sufficient literature review.

Weaknesses:

1. From a technical point of view, it is not clear what the main technical novelties of this paper compare to [ZLJ+20].
2. The paper suffers from a lack of a discussion section at the very end of the paper.
3. In section 3 it would be nice if the authors emphasize more the role of "low dimensions", how they characterize the notion of "low dimensions" and technically exactly at which part they are exploring the low dimension phenomena.

---

> ### Author Response · Authors · 2022-07-31
> **Response to Reviewer 4qyS**
>
> We thank the reviewer for their careful review with pertinent questions about concepts that we can better explain and for catching typos in the text. We would first like to address the major remarks.
>
> ### Novelty compared to Zhang et al
>
> Firstly, we again acknowledge — as we have consistently done in our submission — that the work of Zhang et al is a remarkable breakthrough in the field of non-convex non-smooth optimization. That said, their algorithm (in lines 4 and 14 of Algorithm 1 in <http://proceedings.mlr.press/v119/zhang20p/zhang20p.pdf>) invokes the oracle described by (2) in our paper. We would like to emphasize that the use of this oracle makes the paper of Zhang et al intractable (c.f. Lines 36 - 49; as another example, please also see our response to Reviewer **XcwW** in this regard).
>
> Therefore, **we believe that designing an algorithm with identical provable guarantees but without this intractability is a fundamental question of both theoretical and practical interest: theoretically, we obtain a result with a far weaker assumption, and practically, we make their algorithm tractable, especially in the very setting (deep neural networks) that motivated their paper.** Since we are able to achieve this goal in our paper, we, therefore, view our result as *stronger* than that of Zhang et al.
>
> Our technical novelty involves applying ideas from *randomization to our geometric insights* into Zhang et al’s algorithm.
>
> ### The role of low dimensions in our second contribution
>
> We clarify that the reason we say our algorithm works better in low dimensions is since we have a dependence on the dimension in our second contribution.
>
> We would like to take this space to segue into highlighting our second contribution, which we consider just as interesting and even more technically elegant (we also acknowledge that the significance of this contribution going unnoticed is probably a signal for us to improve our clarity, and we will do so by adding more background on cutting-plane methods, as we also promise **Reviewer hvUX**).
>
> We make the novel observation that a lack of decrease in function value implies that we have, roughly, access to the subgradient oracle of the convex program of finding the minimum norm element in the subdifferential at the current iterate. **This allows us to use the cutting-plane framework to solve this problem, which naturally trades off a $L/\epsilon$ factor for $d \log (L/\epsilon)$, an improvement in the time complexity for $L$-Lipschitz functions (as well as weakly convex functions) in low dimensions**.
>
> This tradeoff between dimension and accuracy comes about naturally since cutting-plane methods, unlike first-order methods, provide high-accuracy guarantees (i.e., with polylogarithmic dependence on $1/\epsilon$ as opposed to polynomial dependence on it) in exchange for a worse dependence on the dimension. (Please also see our remarks in the common response addressed to all the reviewers at the top of the page).
>
> Given the extremely scant structure we impose on our problem (no smoothness, no convexity), we do not view this restriction to low dimensions as a limitation. **Indeed, as was shown in Bubeck and Mikulincer, even the case of $d = 3$ for this problem is very hard, and our result is conjectured optimal based on the work of Bubeck-Mikulincer and Carmon, Duchi, Hinder, and Sidford**.
>
> ### Discussion section
> We thank the reviewer for their feedback on the presentation suggesting a discussion section at the end. We will take this into consideration and will add such a section comprising a condensed version of some of the above relevant responses with some suggestions for future work.
>
> ### Answers to questions in the next comment
> Due to a limit on the number of characters, we are answering the reviewer's specific questions in the next comment.
>
> (1/2)

---

> > ### Author Response · Authors · 2022-07-31
> > **Response to Reviewer 4qyS (2/2)**
> >
> > ### Answers to specific questions
> >
> > > Lines 110 - 111: "Famously, [ZLJ+20] showed that…
> >
> > We thank the reviewer for catching this typo. It was Goldstein who showed this, and we meant to say [Gol77] instead; the fact is also formally stated in Theorem 1 in line 111. We will fix this.
> >
> > > In the proof of Lemma 3, what are the definitions of R+y and Q?
> >
> > $R_+\\{y\\}$ is the nonnegative span of $y$; more precisely, $R_+\\{y\\} := \\{\lambda y \colon \lambda \geq 0\\}$. $\mathcal{Q}$ is the set of directions $y$ such that $f$ is differentiable at almost every point of $\mathbb{R}_+\\{y\\}$. We will add this to our updated manuscript.
> >
> > > Computing the center of gravity
> >
> > We note that $\Omega_k$ is an intersection of a ball and some half spaces, hence we can compute its center of gravity in polynomial time by taking an average of the empirical samples from this convex set. Furthermore, our focus was on minimizing the oracle complexity and have used a simple cutting-plane method for the paper; any algorithm in this class may be used.
> >
> > > Definition of the covariance matrix of a convex set
> >
> > We define the center of gravity of a convex set K as $\mathbb{E}(x)$ for any random point $x \in K$ and its covariance matrix as $\mathbb{E} ((x-\mu)(x-\mu)^\top)$. It can be interpreted as the best ellipsoid approximating the set; the more ``pointy’’ the set is, the higher the condition number of the matrix is.
> >
> > Please do let us know if there are any additional clarifications we can provide. We hope our responses, particularly on the difference from Zhang et al and the significance of our second contribution, elevate your view of our paper. We look forward to engaging with the reviewer in further discussions.
> >
> > (2/2)

---

> > > ### Comment · Reviewer_4qyS · 2022-08-06
> > > **Acknowledging and further suggestions**
> > >
> > > I would like to thank the authors for their complete response and great rebuttal. The authors addressed all of my concerns. According to the rebuttal and other reviews, I increased my score to a 7.

---

> > > > ### Comment · Area_Chair_c29a · 2022-08-08
> > > > **Update of review?**
> > > >
> > > > Dear reviewer,
> > > >
> > > > I wanted to check if you already had the opportunity to update your review.
> > > >
> > > > Best,
> > > > AC

---

> > > > > ### Comment · Reviewer_4qyS · 2022-08-08
> > > > > **Reply**
> > > > >
> > > > > Dear AC,
> > > > >
> > > > > I just updated my review and increased my score to a 7.

---

### Official Review · Reviewer_XcwW · 2022-07-24

**Rating:** 7
**Confidence:** 3
**Soundness:** 3 good
**Presentation:** 3 good
**Contribution:** 2 fair

**Summary:**

This paper deals with optimization in the non smooth and non convex setting. This work is mainly based on the definitions of [Gol77] and recent results of [ZLJ+20].

It allows to find an efficient algorithm (combination of Algorithms 1 and 2) which guarantees (Corollary 5 and Theorem 6) hold only assuming $L$-Lipschitz continuity and almost everywhere differentiability (in the sense of the Lebesgue measure).

Then, authors propose a cutting-plane method to solve the intermediate minNorm problem (previously solve by Algorithm 1).
The advantage of this is the improvement in complexity dependance to $\epsilon$.
The drawback is the dependence on $d$.
This is therefore useful in small dimension.

**Questions:**

l.47: I think what the cited paper means is that if one has access to the solution of (2) for the functions that are composed together, then the chain rule provides the solution for the function of interest.

l.61: This is a strong affirmation based on the assumption that the previous claim is false.

l.110: « Famously, [ZLJ+20] showed ». I think authors here refer to « [Gol77] ».

l.113: minor: authors use the notation $x_0$ to refer to the first iterate. Then use the indices starting from 1 for $g$. I suggest uniforming the notations.

l.113: I also suggest removing the O notation in the last equation since it seems we have $T\leq\frac{\Delta}{\delta \epsilon}$.

l.123: « A short computation shows that this is sure to be the case for all small $\lambda > 0$ as long as $\left< u, g \right> \leq \|g\|^2$ ». As one can see in l.132, this argument relies on the first order expansion on the norm of interest. the second term is non negative, but for small $\lambda$, we expect the first term to be dominant. This is actually true if and only if the first term is not zero, hence the inequality $\left< u, g \right> \leq \|g\|^2$ must be replaced by the strict inequality: $\left< u, g \right> < \|g\|^2$. Reason why the $\delta / 2$ is used in equation (4) and Algorithm 1 even uses $\delta/4$.


**Limitations:**

No limitations.

**Strengths And Weaknesses:**

Strengths:
- Good guarantees with few assumptions.

Weakness:
- [ZLJ+20] already got similar results. Only the need of solution to (2) is assumed. And the claim made by this paper on this oracle seem strong (see first point of next section).

===================================================================================================
========================================= After rebuttal ==============================================
===================================================================================================

The oracle [ZLJ+20] uses is indeed not computationally easy to build, for instance not compatible with the chain rule.

---

> ### Author Response · Authors · 2022-07-31
> **Response to Reviewer XcwW**
>
> We thank the reviewer for their thorough review, especially for their careful checking of all our calculations inside proofs. We address the major concerns first and then answer individual questions.
>
> ### Improvement over Zhang et al
>
> We would first like to address the impression we seem to have conveyed that we recover Zhang et al’s result under merely *slightly weaker* assumptions.
>
> Please note that Zhang et al’s Lemma 2 refers to only computing the directional derivative via the chain rule. So, for instance, for a ReLU network, one is *certainly* able to compute this — we do not claim that this cannot be done. However, owing to how complicated the subdifferential (of the composition of functions representing a ReLU network) can get, solving (2) in our paper — in other words, invoking the oracle of Zhang et al — can be **computationally intractable**. This, in our opinion, makes their oracle non-standard and also their algorithm incomparable to those that do use the standard first-order oracle.
>
> Throughout our paper, we consistently acknowledge the remarkable contribution of Zhang et al, but the above is, in our view, an important limitation of their work. **Our first focus has been to recover their result without their limitation and under only the standard first-order oracle model**. This strengthens our result over Zhang et al’s by making it *actually implementable* in the very settings that motivated this problem and, in our view, constitutes an important contribution to the question of providing non-asymptotic convergence guarantees for non-convex non-smooth functions.
>
> ### Highlighting our second contribution
>
> We would also like to draw the reviewer’s attention to our *second* contribution, which we consider of equal importance and perhaps of even more technical elegance.
>
> We make the novel observation that a lack of decrease in function value implies that we have, roughly, access to the subgradient oracle of the convex program of finding the minimum norm element in the subdifferential at the current iterate. **This allows us to use the cutting-plane framework to solve this problem, which naturally trades off a $L/\epsilon$ factor for $d \log (L/\epsilon)$. We also get improved guarantees for weakly convex functions using this approach**.
>
> We do not view the restriction to low dimensions as a limitation. **Indeed, as was shown in Bubeck and Mikulincer <http://proceedings.mlr.press/v125/bubeck20b/bubeck20b.pdf>, even the case of $d = 3$ for this problem is very hard, and our result is conjectured optimal based on the work of Bubeck-Mikulincer and Carmon, Duchi, Hinder, and Sidford <https://arxiv.org/abs/1710.11606>**.
>
> ### Individual questions
>
> We answer the major questions below and will incorporate fixes to all typos caught (Thank you very much for this!)
>
> - L47, L61. Please see our response above.
> - L110. You are right, this is indeed a typo on our part; it should be [Gol77], and we will fix this. Please see Theorem 1 in l113 where we cite [Gol77] formally stating this.
> - L113. We agree with your observation of our inconsistent notation and will fix this. Thank you! Regarding the bound on T, please note Theorem 6, where $T \leq 4  \frac{\Delta}{\epsilon \delta}$, so we believe that the Big-Oh notation makes sense here.
> - L123. Thank you for carefully checking this: we’ll fix it.
>
>
>
> Please do let us know if there are any additional clarifications we can provide. We hope our responses, particularly on the difference from Zhang et al and the significance of our second contribution, elevate your view of our paper. We look forward to engaging with the reviewer in further discussions.

---

> > ### Comment · Reviewer_XcwW · 2022-08-07
> > **Response to authors**
> >
> > I thank the authors for clarifications.
> >
> > - I agree that authors acknowledge the work of Zhang et al. all along their paper, and that they don't claim contributions that is not theirs.
> > I also agree that their formulation of the oracle is more pleasant.
> >
> > However, I don't think the oracle of Zhang et al. is computationally intractable.
> > Indeed, although I agree that the description of the sub-differential of a RELU neural network is complicated, I think the right way of solving (2) is not to completely describe this sub-differential and then solve (2).
> >
> > I think the chain rule solves (2) automatically assuming we solve (2) for each composite functions which are simple ones.
> >
> > Indeed, assume for instance that $f = f_1 \circ f_2$, where $f_2$ maps $\mathbb{R}^n$ to $\mathbb{R}^m$ and then $f_1$ maps  $\mathbb{R}^m$ to $\mathbb{R}$.
> >
> > Assuming we are able to find easily $g_1 \in \partial f_1(y)$ such that $g_1^T u = f_1'(y, u)$ and we can find $J_2$ a sub-jacobian of $f_2$ in $x$ such that $J_2 u = f_2'(x,  u)$ (Note the latest is precisely finding the $m$ gradients of the components functions of $f_2$ that lead to (2) for each component's directional derivative).
> >
> > Then $J_2^T g_1$ is a sub-gradient of $f_1 \circ f_2$ and is even the one backpropagation provides.
> > And we can verify that from the assumptions on composite functions, we have $g_1 J_2 u = f_1'(f_2(x), g_2 u) = (f_1 \circ f_2)'(x, u)$.
> >
> > Hence we only need to verify each simple functions provide the right sub-gradient, and then the chain rule will provide the right one too.
> >
> > - Concerning the dimension concern, I agree the result is interesting as solution to a hard theoretical problem. The dependency in dimension is a limitation in the practical use of the algorithm. However, this is not a problem to my point of view, as soon as authors thoughtfully decided to present a mostly theoretical problem and that this practical limitation is clearly stated.
> >
> > - About the big-Oh notation: the difference between Th1 and Th6 is the oracle one uses to get the smallest subgradient. In Th1, one has access to the exact one, and in Th6, one only can access an estimation of it obtained with Alg1.
> > Contrary to Th6, Th1 leads to $\min ||g_t|| \leq \varepsilon$ for some $T \leq \frac{\Delta}{\delta \varepsilon}$.
> > Of course the big-Oh notation is not wrong, and the following remark is not major at all, but the authors can rewrite l.113 replacing $T = O\left(\frac{\Delta}{\delta \varepsilon}\right)$ by $T \leq \frac{\Delta}{\delta \varepsilon}$ which is more precise and stronger.
> >
> >
> > **Concluding:**
> > The two last points are not major, just small remarks. I also agree that their oracle is more pleasant than the one in Zhang et al. which is not standard. However, I disagree with the claim that computing it is computationally intractable.

---

> > > ### Author Response · Authors · 2022-08-07
> > > **Chain rule**
> > >
> > > Thank you for your comment. While the strategy you outline is very natural, it seems to be flawed. The proposed computation  relies on the validity of the chain and sum rules both for the directional derivative and the subdifferential. Although the chain rule is indeed valid for the directional derivative of a composition of directionally differentiable functions, the chain rule (and sum rule) can easily fail for the computation of the subdifferential. Consider for example the function $f(x,y)=f_1(x,y)+f_2(x,y)$ with $f_1(x,y)=|x|$ and $f_2(x,y)=-|x|$. Choose the direction $u=(0,1)$ and set $z=(0,0)$. Then $f_1'(z,u)=f_2'(z,u)=0$ and $\partial f_1(z)=\partial f_2(z)=[-1,1]\times \{0\}$. Therefore, according to the proposed construction for $f_1$, we may choose the subgradient $v_1=(-1,0)$ and for $f_2$ we may choose the subgradient $v_2=(-1,0)$ since $<v_1,u>=<v_2,u>=0$. But then $v_1+v_2=(-2,0)$, which is clearly not a subgradient of $f=0$ at $z$. In summary, the proposed strategy may in general fail to yield a valid subgradient of the function $f$.

---

> > > > ### Comment · Reviewer_XcwW · 2022-08-07
> > > > **I still disagree**
> > > >
> > > > Dear authors,
> > > >
> > > > first, I guess in the example you proposed $\partial f_i (x)$ refers to $\partial f_i (0)$ as 0 is the only point of non differentiability.
> > > >
> > > > Then, you are right saying $\partial f_1(0)+\partial f_2(0)=[-2,2]$. But, precisely the sum rule (which is a particular case of the chain rule) will precisely return 0.
> > > > Indeed, with $<g_1, u> = f_1'(0, u) = |u|$ and  $<g_2, u> = f_2'(0, u) = -|u|$, then $g_1 + g_2 = 0$ as we want.
> > > >
> > > > Then this example shows how the chain rule (and here the sum rule) automatically finds the right sub-gradient as soon as the oracle knows and returns the right sub-gradient for composite functions.
> > > >
> > > > Best regards.

---

> > > > > ### Author Response · Authors · 2022-08-07
> > > > > **Chain rule fails**
> > > > >
> > > > > There seems to have been a time lag in the response. The updated response is below. Let us know your thoughts.
> > > > >
> > > > > "Thank you for your comment. While the strategy you outline is very natural, it seems to be flawed. The proposed computation  relies on the validity of the chain and sum rules both for the directional derivative and the subdifferential. Although the chain rule is indeed valid for the directional derivative of a composition of directionally differentiable functions, the chain rule (and sum rule) can easily fail for the computation of the subdifferential. Consider for example the function $f(x,y)=f_1(x,y)+f_2(x,y)$ with $f_1(x,y)=|x|$ and $f_2(x,y)=-|x|$. Choose the direction $u=(0,1)$ and set $z=(0,0)$. Then $f_1'(z,u)=f_2'(z,u)=0$ and $\partial f_1(z)=\partial f_2(z)=[-1,1]\times \{0\}$. Therefore, according to the proposed construction for $f_1$, we may choose the subgradient $v_1=(-1,0)$ and for $f_2$ we may choose the subgradient $v_2=(-1,0)$ since $<v_1,u>=<v_2,u>=0$. But then $v_1+v_2=(-2,0)$, which is clearly not a subgradient of $f=0$ at $z$. In summary, the proposed strategy may in general fail to yield a valid subgradient of the function $f$."

---

> > > > > > ### Comment · Reviewer_XcwW · 2022-08-07
> > > > > > **Convinced**
> > > > > >
> > > > > > This is indeed a convincing example!
> > > > > > Clarke sub-differential does not exhibit good properties.
> > > > > >
> > > > > > This indeed motivates this work even more.
> > > > > > I will upgrade my score accordingly.

---

### Author Response · Authors · 2022-07-31
**Overall summary with some clarifications (based on reviews)**

We are deeply grateful to all the reviewers for such well-thought-out reviews of our submission, their appreciation of our ideas, and their multiple constructive efforts in helping us strengthen our work. We will address all their major questions and remarks by responding directly to each review (and will incorporate all their suggestions), but first here at a top level, we would like to briefly reiterate our two key contributions.

### First Contribution

With the advent of deep learning, the question of studying complexity guarantees of algorithms for optimizing non-smooth non-convex functions (of which modern neural networks are a classic example) has become one of paramount importance. The elegant result of Zhang et al (ZLJSJ2020, <http://proceedings.mlr.press/v119/zhang20p.html>)  is the first in a long line of work to provide a non-asymptotic (i.e., finite-iteration) convergence guarantee for finding a stationary point for some meaningful notion of stationarity (also introduced by Zhang et al).

While a significant breakthrough in both result and technique, one crucial limitation of the algorithm of Zhang et al is that it invokes a rather strong (and non-standard) oracle (c.f. Lines 36 - 49). we provide answers to specific questions of reviewers in this regard in our direct responses to them).

**Our first major contribution is a simple modification of Zhang et al’s algorithm yielding their result under the standard first-order oracle model**. This result is a *much stronger* guarantee than that of Zhang et al, since our algorithm (Algorithm 1 and 2), unlike Zhang et al’s, is *actually implementable* in the very settings that motivated this problem.

We obtain our result by applying ideas from *randomization to our geometric insights* into Zhang et al’s algorithm. Specifically, our Algorithm 2 returns a $(\delta, \epsilon)$-stationary point of $f$ by moving along the direction of the negative normalized  $g_k$, where $g_k$ is either a descent direction or a minimum norm element of the subdifferential of the current iterate; the computation of $g_k$ in Algorithm 1 is our key algorithmic contribution.

We hope the simplicity of our approach does not bely the significance of our result on this front.

### Second Contribution

**Our second contribution is to improve, in low dimensions, the iteration complexity of the task of computing the minimum norm element of the subdifferential at a point**. Specifically, Algorithm 1 converges in $O(\frac{L^2}{\epsilon^2})$ iterations, whereas Algorithm 3 takes $O(\frac{Ld}{\epsilon} \log (1/\epsilon))$ iterations for $L$-Lipschitz functions and $O(d \log(L/\epsilon) \log(\delta \rho/\epsilon))$ iterations for $\rho$-weakly-convex functions.

Our key insight is that *if the function does not decrease in a candidate descent direction, then the guarantee in this direction ((8), Definition 2) is roughly equivalent to the first-order optimality on the minimum-norm element problem we seek to solve* (c.f. Lines 175 - 188), which is precisely the requirement for solving this optimization problem via a cutting-plane method. (Per the feedback from **Reviewer hvUX**, we plan to add a short background section on cutting-plane methods in the appendix.)

Proving this insight (Lemmas 9 and 10) requires a simple, yet novel, perspective on the minimum norm element problem, and this insight itself could have, in our opinion, broad applications in optimization.

In addition to this technique, the result is itself, in our opinion, significant in its own right, as we describe in Lines 76-90. Specifically, based on a prior conjecture by Bubeck and Mikulincer and the work of Carmon, Duchi, Hinder, and Sidford, **we believe that our result is optimal for low dimensions. Obtaining such a result for even dimension $d = 3$ is considered a hard problem (c.f. Bubeck and Mikulincer)**.

### Closing Remarks

Overall, therefore, we believe our contribution comprises results stronger than prior work, conjectured optimal in certain settings, with novel techniques, all applied to a central problem in modern machine learning.

---

### Comment · Area_Chair_c29a · 2022-08-03
**Discussion period**

Thanks to all reviewers and authors for their work on this submission.

As the discussion period starts, I want to make sure that reviewers have read the author's response, and if needed react to it.

This can be done either by communicating with authors or in private conversation within the reviewing team.

Reviewer 4qyS : Has the author's response adressed your comments?

---

### Meta-Review · Area_Chair_c29a · 2022-08-23

**Recommendation:** Accept
**Confidence:** Certain

**Metareview:**

The main contribution of this work is to extend and improve previous results regarding optimization complexity of Lipschitz functions.

In particular, it addresses issues in previously proposed algorithms that did not make it implementable. The newly proposed algorithm does not use the strong oracle that was required in previous work, and only uses first-order information, as is common in large-scale optimization problems in ML. There is also improved analysis of the complexity of the algorithms.

The reviewers had many questions for the authors, and there was a fruitful and constructive discussion between all parties. Their remarks have been adressed, and all reviewers recommend to accept this work, and so do I.

The comments that I would personally have is that 1) For an implementable algorithm, an implementation would have been nice 2) Since the motivation is to train models in an ML setting, some discussion about the "quality" of these stationary points from the point of view of generalization abilities would also complete nicely this work.

**Award:**

No

---

### Decision · Program_Chairs · 2022-09-14

Accept